# Systematic Review on Tumor Microenvironment in Glial Neoplasm: From Understanding Pathogenesis to Future Therapeutic Perspectives

**DOI:** 10.3390/ijms23084166

**Published:** 2022-04-09

**Authors:** Andrea Bianconi, Gelsomina Aruta, Francesca Rizzo, Luca Francesco Salvati, Pietro Zeppa, Diego Garbossa, Fabio Cofano

**Affiliations:** 1Department of Neuroscience “Rita Levi Montalcini”, University of Turin, 10126 Turin, Italy; gel.aruta@gmail.com (G.A.); rizzofrancesca839@gmail.com (F.R.); pietro_zeppa@yahoo.it (P.Z.); dgarbossa@unito.it (D.G.); fabio.cofano@gmail.com (F.C.); 2Unit of Neurosurgery, Santa Corona Hospital, 17027 Pietra Ligure, Italy; diksalvati@gmail.com; 3Spine Surgery Unit, Humanitas Gradeningo, 10100 Turin, Italy

**Keywords:** glioma, microenvironment, GAMs, microglia, extracellular vesicles

## Abstract

Despite the multidisciplinary management in the treatment of glioblastomas, the average survival of GBM patients is still 15 months. In recent years, molecular biomarkers have gained more and more importance both in the diagnosis and therapy of glial tumors. At the same time, it has become clear that non neoplastic cells, which constitute about 30% of glioma mass, dramatically influence tumor growth, spread, and recurrence. This is the main reason why, in recent years, scientific research has been focused on understanding the function and the composition of tumor microenvironment and its role in gliomagenesis and recurrence. The aim of this review is to summarize the most recent discovery about resident microglia, tumor-associated macrophages, lymphocytes, and the role of extracellular vesicles and their bijective interaction with glioma cells. Moreover, we reported the most recent updates about new therapeutic strategies targeting immune system receptors and soluble factors. Understanding how glioma cells interact with non-neoplastic cells in tumor microenvironment is an essential step to comprehend mechanisms at the base of disease progression and to find new therapeutic strategies for GBM patients. However, no significant results have yet been obtained in studies targeting single molecules/pathways; considering the complex microenvironment, it is likely that only by using multiple therapeutic agents acting on multiple molecular targets can significant results be achieved.

## 1. Introduction

Despite the multidisciplinary management in the treatment of glioblastoma, the average survival is 15 months [1]. In recent years, molecular biomarkers have gained more and more importance both in the diagnosis and therapy of glial tumors [2]. Non neoplastic cells are believed to constitute up to 30% of glioma mass and are responsible for the creation of the so-called tumor microenvironment (TME) [3,4], which includes specialized glioma stem cells (GSCs), glioma cells, stromal cells including resident glial cells and immune cells such as monocytes, tumor-associated macrophages/microglia (TAMs), and lymphocytes [5]. Intra- and inter-tumor heterogeneity is therefore mainly related to the peritumor microenvironment. Understanding which type of cell constitutes the majority of TME, what their role is, and to what extent they can influence the development and spread of gliomas assumes great importance, especially for therapeutic implications, in order to focus pharmacological research on a specific cellular target. TME dramatically influences tumor growth, spread, and recurrence; for example, TAMS acts directly to promote glioblastoma growth and to create an immunosuppressive TME. Recent evidence indicates how cancer cells can subvert the peritumor microenvironment, promoting angiogenesis, invasion, and metastasis [6]. Understanding how glioma cells interact with cells in the microenvironment is an essential step in understanding disease progression.

The activation of microglial cells and the recruitment of circulating monocytes in the brain is mainly induced by chemokines, neurotransmitters, ligands of complement receptors, and a new mode of communication by the release of extracellular vesicles (EVs). Considering the large number of studies present in the literature regarding the role of the tumor microenvironment in gliomagenesis, the goal of our study is to summarize the current evidence concerning the TME, better understanding TAM composition, and highlighting the mechanisms of interaction between microglia and tumor cells, as this could have implications both for therapeutic and prognostic aims. We focus, on one hand, on the current understanding of pathogenesis of tumor growth and progression, and on the other hand on how these mechanisms could have a clinically significant impact in the coming years, highlighting the most promising therapeutic prospects on this topic, both on mouse and in vitro models.

## 2. Materials and Methods

A systematic review according to PRISMA-P (Preferred Reporting Items for Systematic review and Meta-Analysis Protocols) guidelines was performed (Figure 1). The study protocol was registered, and the systematic search strategy of the review is documented online at the International Prospective Register of Systematic Reviews Registry (PROSPERO). We used an online databases search using PubMed, EMBASE, and Google Scholar applying the following research terms used as free terms, keywords, or MeSH terms: “glioma”, “glioblastoma”, “Glioma stem cells”, “TAMs”, “GAMs”, “microglia”, “miRNA”, “extracellular vesicles”, “microenvironment”, “progression”, “growth”, and combining them with AND, OR, or NOT operators. We considered only English language papers published in the last 20 years. Using the search terms mentioned above, the literature search retrieved 482 articles. Based on the title and the abstract, a first screening of the resulting 482 articles was carried out, and irrelevant studies were excluded. Articles regarding metastases, extra cerebral neoplasms, or other types of brain cancer were not considered. Case reports, letters to the editor, retrospective cohort studies, and non-systematic reviews were excluded, while prospective studies, case–control, cohort or randomized trials, and systematic reviews were included. Considering the relative novelty of the topic under consideration, studies involving mice models or in vitro models were also considered, particularly considering the therapeutic perspectives. After identification of relevant works, additional articles were identified by crossing references. A total of 53 full-text articles were assessed for eligibility and were independently screened by the first four authors. Disagreements were resolved through consensus by discussion. Each paper’s full text was read and critically assessed by the first four authors. Finally, 13 research articles and 16 reviews were selected.

Considering the width of the topic under consideration, for ease of interpretation of the results, the studies analyzed were divided between studies that considered molecular pathogenesis and tumor progression mechanisms, differentiated from those about therapeutic perspectives. Regarding the reviews, we considered only systematic reviews and/or meta-analyses where the research methodology was clearly specified. Studies that considered roles of microglia, EVs, or miRNAs beyond gliomas were excluded, even if reviews that dealt with other cerebral tumors, or even extra cerebral neoplasms including gliomas among them were included, and the data exclusively related to gliomas were extracted. Only the articles clearly stating the effect of a single or multiple molecules directly on tumor progression were included, thus excluding articles concerning the signaling of cellular pathways involved in other mechanisms. The articles concerning extracellular vesicles, and those concerning the other aspects of microglia were separated in different subheadings. Finally, the articles, experimental or reviews, concerning the specific therapeutic possibilities connected to the modulation of the peritumor microenvironment were considered separately.

## 3. Results and Discussion

### 3.1. Differentiation and Role of Resident Microglia and Circulating Monocytes

In human gliomas, TME is constituted by a mix of two different cell types: microglia, the resident macrophages of CNS, and monocytes, which originate from hematopoietic stem cells (HSCs) and travel through the blood to reach brain when blood–brain barrier (BBB) is damaged. These cells not only have two ontogenetically different origins, but also express some different surface receptors and have, perhaps, different mechanisms of action in supporting gliomas growth and spread. (Figure 2).

Normal brain tissue is colonized exclusively by quiescent microglia. In the context of microglia cells differentiation in normal brain tissue, a recent study demonstrated the existence of different microglia subsets by using a single-cell-based immune phenotyping by time-of-flight mass cytometry (CyTOF) and single-cell RNA sequencing (scRNA-seq) [7]. Conducted on 4396 microglial cells from control brain regions of 15 adults, in this study, the authors identified eight major clusters (C1–C9, with the exclusion of C4) within the microglia cloud. C2 subset was characterized by the strong expression of major histocompatibility complex II (MHC-II) and antiviral immunity genes, such as HLA-DRA, CD74, and IFI44L; C3 subset was associated with the high expression of CX3CR1 and TMEM119; C6 and C7 subsets were characterized by low expression of CX3CR1 and high expression of integrin-receptor-binding protein and metabolism genes, including SPP1, APOE, and LPL. These clusters are therefore involved in antigen processing and presentation of peptide antigen. The remaining smaller clusters C1, C5, C8, and C9, associated with the expression of chemokine and cytokine genes, including CCL2 and IL1B, are instead involved in positive regulation of chemotaxis and MAPK cascade. In the same study, an association between brain regions and specific microglia clusters was demonstrated. In fact, C2 and C5–C7 were overexpressed in white matter microglia, while C3 and C8 were overexpressed in gray matter microglia [7]. Under pathological conditions, microglia density increased through active recruitment of bone marrow-derived microglial progenitors from the bloodstream and, at the same time, HSCs infiltrate brain through chemoattraction [8,9,10].

The distinction between microglia and invading monocytes has always been made using anti-CD45 antibody, with microglial cells expressing low CD45 (CD45^low^) and HSCs with a high expression of CD45 (CD45^high^). Different studies, based on sampling human glioma by fluorescent-activated cell sorting, demonstrated that the majority of glioma associated macrophages and monocytes (GAM) are actually HSCs. A more recent study, instead based on irradiation chimeras, showed that resident microglia is the most represented member of GAM [11]. Another study, based exclusively on CX3CR1 or CCR2 antibodies, demonstrated again the prevalence of HSCs in TAM (CCR2+ CX3C1−) and the relatively low presence of resident macrophages (CCR2− CX3C1+) [12]. In 2020, Friebel et al., by using a combination of experimental approach, successfully identified the origin of each TAM population stemming either from CNS-resident microglia or HSCs. They confirmed that glioma TME is mostly characterized by TAM of microglial origin, but, interestingly enough, they also found a major invasion of HSCs in patient with IDH1 wild type gliomas in comparison to IDH mutated gliomas [13].

The lack of consensus in the literature, at least partly determined by the different methods of cell sorting (surface antibody or chimeras), underlines the necessity of new organized studies, in order to better understand TAM composition and to assess if targeting these cells can have a prognostic significance in gliomas treatment.

### 3.2. Microglia Recruitment Factors

Tumorigenesis is not only driven by genetic alterations and function of tumoral cells, but is also highly influenced by the tumor microenvironment, composed by cellular and non-cellular elements. In brain tumors, especially gliomas, tumor stroma is mainly composed of immune cells such as resident microglia and circulating macrophages and monocytes.

The activation of microglial cells and the recruitment of circulating monocytes (HSCs) in the brain is mainly induced by chemokines, neurotransmitters, and ligands of complement receptors.

One of the first chemoattractant identified was monocyte chemoattractant protein-1 (MCP-1), also known as CCL2. MCP-1 role in tumor growth has been studied in rats, where tumors generated with CCL2-expressing cells were more than three-fold larger in size than control-transfected tumors [14]. Additionally, it has been observed that in vitro chemoattraction could successfully be inhibited by an MCP-1-neutralizing antibody, while in an experiment conducted in vivo in mice, glioma cell-derived MCP-1 increased GAM infiltration. MCP-1 expression has also been associated with glioma grade [14], neoangiogenesis, tumor cells proliferation, and invasiveness [15].

However, a more recent study enhanced the importance of monocyte chemotactic protein 3 (MCP-3) role in microglia and HSCs recruitment over MCP-1 in human gliomas [16].

In a murine astrocytoma model, tumor secreted stroma-derived factor-1 (SDF-1) has been found responsible for microglia and macrophage recruitment, especially in hypoxic areas of brain [17].

Chemokine CX3CL1 (also known as fractalkine or neurotactin) exists as a membrane-anchored protein and as a potentially soluble form. The highly specific receptor for CX3CL1, called CX3CR1, is expressed by microglial cells in normal brain tissue by endothelial cells and platelets [18]. In a study conducted on GAM, CX3CR1 was overexpressed at both mRNA and protein level, and it was found that CX3CL1 promoted their adhesion and migration in vitro, through the induction of matrix metalloproteases (MMPs) 2, 9, and 14 [19]. Conversely, a more recent study demonstrated that loss of the fractalkine receptor CX3CR1 in microglia and monocytes in glioma-bearing mice resulted in increased tumor incidence and shorter survival, but did not affect the accumulation of microglia/macrophages in peri-tumoral areas. According to the authors, the first effect is related to the upregulation of interleukin 1β (IL-1β) expression in CX3CR1-deficient microglia macrophages and monocytes. IL-1β, indeed, has been found responsible for tumor growth and infiltration of cancer stem cells phenotype, both in mice and human Proneural glioma stem cells [20].

These discordant data suggest that the role of fractalkine in gliomagenesis and its therapeutic implications has yet to be defined.

Growth factor glial cell-derived neurotrophic factor (GDNF), secreted by murin and human glioma cells, is a strong microglia attractant in vitro, while having a small role in glioma-induced astrogliosis [21].

CSF-1, another soluble factor released by glioma cells, can serve as chemoattractant in vivo. In mice treated with CSF-1R antagonist, in fact, a reduction in GAM density and of glioblastoma invasiveness was observed [22]. CSF-1 is also responsible for microglia polarization into the pro-tumorigenic M2-like phenotype [23].

The knockdown of granulocyte -macrophage colony-stimulating factor (GM-CSF) in brain slices reduced microglia-macrophages invasion and the growth of intracranial gliomas in vivo [24].

Epidermal growth factor (EGF), whose receptor (EGFR) is expressed on microglial cells, is responsible for GAM migration and motility in in vitro experiments in a dose-dependent way. Moreover, as EGF is secreted by activated microglia itself in vivo, it may act as an autocrine modulator and amplifier of microglial cells functions.

In conclusion, several soluble factors, secreted by both glioma and microglial cells, can act as GAM chemoattractant in glioma bearing brain. Therefore, the inhibition of these ligands and/or their receptors could represent a new frontier in the research for therapeutic strategies in glioma patients.

### 3.3. The M1–M2 Polarization of GAM

Different pathological stimuli can activate quiescent resident microglia, such as injuries in brain trauma, tumors, and viral and bacterial infections [25]. Immunogenic antigens such as lipopolysaccharide (LPS) use several receptors, including Toll-like receptors (TLRs), nucleotide-binding oligomerization domain-(NOD)-like receptors, and scavenger receptors (SRs), to induce pathogens clearance through phagocytosis and, in the end, to trigger an immune response inside the central nervous system (CNS) [26]. Via signal transducer and activator of transcription (STAT-1), LPS can also activate the expression of pro-inflammatory cytokines, such as IL-1α, IL-6, IL-12, and IL-23, and chemokines, such as CC-chemokine ligand (CCL)2–5 and CCL8–11. Redox molecules, as NADPH-ossidase and inducible nitric oxide synthase (iNOS) for nitric oxide production, are also involved in the creation of a metabolic state called pro-inflammatory M1-phenotype [27,28].

Microglial cells expand and amplificate this activation by expressing co-stimulatory molecules (CD 40, CD80, and CD86) and exposing high levels of major histocompatibility complex (MHC) II molecules on their surface [27,29,30], serving as antigen presenting cells (APC). Periphery monocytes can be activated by resident microglia through tumor necrosis factor (TNF)-α.

On the other side, an anti-inflammatory M2-like phenotype is simultaneously activated in order to prevent tissue damages, to downregulate immune response, and to lead to CNS recovery.

Different molecules have been identified as M2 polarized microglia/macrophages markers, such as CD163 (scavenger receptor), CD204, and CD206 (macrophage mannose receptor) [31,32].

A further distinction inside M2 polarized cells has been proposed, with three categories, M2a, M2b, and M2c, that are believed to develop in different conditions and to be triggered by various environmental stimuli [33]. These categories exert distinct functions in the contest of anti-inflammatory response.

T-helper cell secretion of IL-4 is responsible for triggering a M2a polarization. The activation of IL4-Receptor (IL-4R) on microglia-macrophages surface, indeed, induces the production, via STAT 6 pathway, of anti-inflammatory cytokines and chemokines such as transforming growth factor (TGF)-β and CCL15, -17, -22, and -24 [34,35]. Simultaneously, IL-4R signaling leads to a silencing of the M1-characteristic NF-κB signaling [27].

M2b subtype is responsible for secretion of IL-1, IL-6, IL-10, CCL1, and TNF-α, via TLRs or IL-1R antagonists [36]. M2b activation leads, eventually, to immune regulation through TH2 and regulatory T cell (Treg) activation.

M2a and M2b subtypes play similar roles in immune regulation: the first can stimulate Th2 response and induce pathogens killing and allergies development, while the second is involved in Th2 response and immune regulation.

M2c subtype, instead, is a major promoter of tumor growth as, in neoplastic conditions, it is responsible for the attenuation of inflammatory responses, tissue remodeling, and matrix deposition [36].

Tumor cells are responsible for production and secretion of IL-10, TGF-β, and glucocorticoids; in particular, IL-10, through STAT3 signal, leads to a rise in TGF-β and peroxisome proliferator-activated receptor (PPAR)-γ expression and, in the end, to the activation of anti-inflammatory response [27].

M2c phenotype facilitates extracellular matrix (ECM) deposition and tissue remodeling by the expression of versican, antitrypsin, and pentraxin 3 [37,38].

For more than twenty years, M2-polarization of TAM has been a cornerstone of our solid tumors knowledge [39,40]. The immunosuppressive environment, also responsible for tissue remodeling and angiogenesis, has been linked to tumor progression [36] and poor prognosis [41]. This paradigm, however, has been challenged in the last few years, especially in brain tumors, where the role of GAM polarization and the immune system in tumor development, spread, and recurrence seems to be more complicated than previously thought [20,42,43].

The first major difficulty is linked to the scarceness of distinct markers for M1 and M2 polarized cells. Surface receptors such as CD68 and cytoplasmic protein IBA1 are considered general microglia/macrophages markers (pan-M/M markers) [32].

Markers such as CD163, CD204, CD206, arginase 1 (ARG1), FIZZ1, and phosphorylated STAT3 (pSTAT3) have been attributed to M2-polarized GAMs, while CD40, CD74, and MHC II have been linked to M1-polarized GAMs [27].

Komohara et al. demonstrated that glioma cell-derived factors, such as TGF-β and macrophage colony-stimulating factor (M-CSF), promote the upregulation of several M2 markers, such as CD163 and CD204, and therefore M2-polarization of GAM [40].

Different studies have been focused on the correlation between survival in patients with high grade glioma and the concomitant expression of either M1 or M2 polarization markers in GAM.

In one study, the analysis of GAM composition showed the predominance of M2 polarization, with a major expression of CD163+ and CD204+ cells, and its association with a worse prognosis [40].

In another study, CD74, a M1-polarization marker, was found to be expressed in GAM and was positively related to longer patient survival [44].

However, this mutually exclusive state of polarization of GAM is true in vitro, but it is now clear that this model cannot represent the complexity of in vivo setting, as different studies demonstrated that GAM can express both M1 and M2 markers at the same time [45,46,47]. Moreover, with the secretion of a variety of soluble factors, glioma cells can induce both M1 and M2 polarization in GAM. Glioma-derived M-CSF and TGF-beta shift microglia and macrophages phenotype towards a M2-type and are responsible for tumor growth [23,40]. In the same way, glioma cells can inhibit M1 phenotype with the activation of mTor and CSF-1 [48]. Other studies have suggested that M1 specific markers or associated pathways positively correlate with glioma growth, such as glioma-derived IL1-β [20].

A more recent study, focused on the prognostic significance of M1/M2 polarization GAM, demonstrated a mixed polarization phenotype, with parallel expression of presumptive M1 and M2 markers, without an univocal state of polarization.

As already mentioned before, Sankowski et al. demonstrated that SPP1 gene, encoding for a proinflammatory cytokine called osteopontine, is more expressed in C6 and C7 clusters of normal brain microglia, especially in people aged over 50 years. Afterwards, they compared osteopontine expression in GBM samples to control the white matter of four subjects >50 years of age, and found that the percentage of IBA1+SPP1+ cells was more than double in GBM samples compared with controls, demonstrating an over-expression of SPP1 gene in the GAM clusters and the similarity of GAM phenotype with the one of aging-associated microglia [7].

In the same study, with the application of the StemID2 algorithm, a gradual downregulation of microglial homeostatic genes (including CX3CR1 and CSF1R) and a concomitant upregulation of inflammatory and metabolism genes (including TREM2 and APOE) was observed in glioma associated microglia.

In conclusion, Sankowski et al. demonstrated the heterogeneity of GAM and its time-related progression from an homeostatic phenotype to a proinflammatory and metabolic phenotype, with lower expression of the microglial core signature and higher expression of proinflammatory genes such as SPP1 and several type I interferon genes, including APOE and CD163, complicating the picture of GAM expression in human glioma even further [7].

Even though the dual M1/M2 model is an oversimplification of in vivo GAM activation, it is still a useful model to find therapeutic target and to direct further tumor environment characterization.

### 3.4. Microglia Regulation of Glioma Growth, Progression, and Angiogenesis

The evidence that microglia accounts for 30% of glioma mass has raised the question about its role in tumor growth and progression.

In 2002, an in vitro study evidenced that the motility of murine glioma cells, when in the presence of microglia, increased by three times [49], while endothelial cells and oligodendrocytes had little effect on glioma cells’ motility. In recent years, different soluble factors released by microglial cells have been identified as glioma growth and motility promoters [50]. TGF-β promotes glioma cells motility with processes that involve integrin and matrix-metalloproteinase induction [51]. In particular, TGF-β superfamily members 1–3 have been identified as immunosuppressive cytokines, which are upregulated in glioma tissues and secreted by tumoral cells [52]. Furthermore, through the activation of matrix-metalloproteinase (MMP)-2 and the suppression of tissue inhibitor of metalloproteinases (TIMP)-2, which degrades extracellular matrix, GAM-secreted TGF-β facilitates glioma cells motility in vitro [51]. To further confirm the importance of MMP-2 role in gliomas spread, a significant association between MMP-2 expression with astrocytomas aggressiveness and unfavorable prognosis has also been demonstrated [53].

To activate pro-MMP2, an inactive pro-form of MMP2, membrane-type1 matrix metalloproteinase 1 (MT1-MMP) is needed. Normal microglia do not normally elaborate MT1-MMP, while in glioma associated microglia MT1-MMP is overexpressed and its expression increases with glioma grade [54].

Toll-like receptor (TLR) 2 has been identified as an important MT1-MMP up-regulation promoter in microglia. There is an increasing list of TLRs endogenous ligand. For example, versican (or CSPG2) is a ECM component, member of chondroitin sulfate proteoglycans family, and it exists in four different iso-forms V0, V1, V2, and V3 (ref). V1 in the brain is responsible for neuronal differentiation and promotes neurite outgrowth, while V2 is an inhibitor of axonal growth [55,56]. In a study, Hu et al. demonstrated that versican V0/V1 was highly expressed by mouse and human gliomas compared with normal brain tissues. Moreover, versican activated macrophage through TLR2, while V1 iso-form significantly induced microglial MT1-MMP expression [37].

As further proof of TLR2 and MT1-MMP importance in glioma growth and progression, a study demonstrated that, in mice with deletion of TLR2, implanted gliomas were significantly smaller and mice survival prolonged compared with wild-type control mice. Furthermore, TLR2-neutralizing antibodies reduced microglial MT1-MMP expression and attenuated glioma growth [57].

The co-chaperone stress-inducible protein 1 (STI-1), a cellular prion protein ligand, synthesized and released by microglial cells, induces glioma cells migration and proliferation in vitro and in vivo [58].

As already mentioned before, CCL2 and its receptor CCR2, expressed on microglial cells surface, can induce IL-6 secretion and intensify glioma invasiveness [15].

Different studies demonstrated that microglia can induce angiogenesis by secreting several pro-angiogenic factors, such as vascular endothelial growth factor (VEGF) [59] and IL-6 [60]. (Figure 2) The administration of Sunitinib, a VEGFR inhibitor, combined with Bevacizumab, a VEGF inhibitor, reduced the infiltration of myeloid cells in gliomas and their vascularization [61]. IL-6 expression in microglial cells is mainly induced by the activation of the receptor for advanced glycation end products (RAGE) [62]. In a study, genetic depletion of RAGE in murine microglia abrogated angiogenesis by downregulating the expression of proangiogenic factors and was associated with animal prolonged survival [63].

All these evidences support the idea that genesis, growth, and progression of gliomas are not exclusively dependent on the intrinsic mutations of neoplastic cells, but are strongly influenced by the surrounding microenvironment and by the communication between tumoral and non-tumoral cells. Hence, it appears essential to strengthen the focus of scientific research on the study and the understanding of GAM in order to identify new therapeutic targets and treatment strategies for glioma patients, which are desperately needed.

### 3.5. Role of Extracellular Vesicles

A critical feature that contributes to poor prognosis is the ability of glioma cells to rapidly and aggressively infiltrate into the surrounding normal brain parenchyma [64]. Intra- and inter-tumor heterogeneity is related to the peritumor microenvironment TME which includes specialized glioma stem cells GSCs, glioma cells, stromal cells including resident glial cells (oligodendrocytes, astrocytes, ependymal cells, and microglia), and immune cells such as monocytes, tumor-associated macrophages/microglia TAMs, and lymphocytes. TAMS acts directly to promote glioblastoma growth and to create an immunosuppressive TME [5]. Recent evidence indicates how cancer cells can subvert the peritumor microenvironment, promoting angiogenesis, invasion, and metastasis [6]. These cells use different communication pathways that facilitate tumor progression. They include direct cell interactions through membrane receptors and their ligands, and the release of soluble factors, such as cytokines, chemokines, and metabolites [65]. (Figure 3) In recent studies, a new mode of communication has emerged, through the release of extracellular vesicles EVs, which carry functional proteomic and genomic information, responsible for altering expression of tumor promoting and tumor suppressing genes, contributing to the disease progression via mediating drug resistance and epithelial-mesenchymal transition [65]. EVs are highly heterogeneous secreted membran bound vesicles that encompass both microvesicles and exosomes. They are naturally released from the cell in numerous biological fluids and extracellular space and also able to cross the blood–brain barrier. The EVs should be described by their physical characteristics such as size and density, biochemical composition, biogenesis, or description of cell of origin. In particular, exosomes are derived from multivesicular bodies, while microvesicles and apoptotic bodies are from plasma membranes and further exosomes size range is 50–100 nm, microvesicles size range is 100–1000 nm, and apoptotic bodies range from 1 to 5 μm. An important source of EVs in the tumor is the small population of glioma stem cells GSCs that are playing a significant role in GBM progression, by EV-mediated transfer of Notch1 protein that is highly enriched in their EVs, or by transfer of the proangiogenic and immunosuppressive factor VEGF-A [64]. Understanding how glioma cells interact with cells in the microenvironment is an essential step in understanding disease progression. According to several studies, the migration of tumor cells is supported by an attraction by the microglia and TAMs through the release of cytokines, chemokines, and proteins of the extracellular matrix, making this cell population approximately one third of the total mass [5]. It is suggested that microglia have downregulated expression of genes involved in the detection of cancer cells and tumor-derived warning signals, as well as genes used to kill the tumor. In contrast, expression of genes involved in facilitating tumor spread are upregulated. These changes appear to be in part EV-mediated [66]. Krapež et al. suggested that drugs encapsulated in the EVs induced the apoptosis of the microglial cells and significantly delayed brain tumor growth [67]. Once the tumor cells release the EVs, the tumor-associated macrophages and the microglia pick them up, and the M1–M2 protumor switch, and the secretion of metalloprotease, IL6, and VEGF are induced [68,69]. The release of EVs from the hypoxic zones of GBM tumors was shown to induce M2 macrophage polarization [65]. A recent study demonstrated the functional effects of miR-21 containing glioma EVs in microglial M2 phenotypic transition in vivo [70]. Glioma EVs were also shown to suppress the activity of natural killer cells and increase the activity of myeloid derived suppressor cells MDSC [70]. These data identify specific changes in the transcriptome of microglia in the presence of glioblastoma that support tumor growth [66]. Neuronal Stem Cells NSC-derived EVs have been studied for their ability to modulate microglia activity [71]. Several inducers of microglial EV release have been found, including Wnt3a, serotonin, and lipopolysaccharide LPS, which also induce an increase in inflammatory cytokines (TNF α and IL-6) in the EV cargo. Given the importance of microglia and infiltrating peripheral macrophages in facilitating glioma cell proliferation and migration, it seems likely that the secretion of EVs by these immune cells would have biological consequences. In the study conducted by Sybren L. N. Maas et al., we are shown transcripts encoding proteins that directly mediate microglia–glioblastoma cellular interactions, such as sialic acid-binding immunoglobulin-like lectin-H that is a microglial sensor of glioblastoma cells. A second group of transcripts that is downregulated in microglia includes those encoding proteins that sense metabolic products potentially released by glioblastoma cells. These transcripts include Gpr183, Adora3, Il6Ra, Cx3cr1, P2ry12, P2ry13, Csf1r, and Csf3r. GPR183 is a sensor for oxysterols, which are released by glioblastoma cells and play a role in recruitment of immune cells. The role of microglia in maintaining brain homeostasis is expressed also by metalloproteases MMPs. MMP enzymes could also play an important role in promoting tumor growth by making space for tumor cells to migrate, invade, and proliferate. Tumor microglial cells are an important source of MMP14 [66]. Microglia in the TME shows an increased expression of several phagocytic receptors, while either maintaining or increasing expression of extracellular matrix degrading enzymes [5]. Microglia avidly took up GBM-EVs [72]. Van der Vos et al. were the first who suggest the cross-talk between GBM and microglial cells. Maire et al. demonstrated that EV-associated DNA represents an unexplored source of information on methylation profiling allowing to classify tumors with unknown genetic alterations and to evaluate changes in the genetic or epigenetic tumor [73]. The experiment of Kristan E. van der Vos et al., for example, showed that high levels of miR-451/miR-21 in GBM-EVs were transferred to microglia with a decrease in the miR-451/miR-21 target c-Myc mRNA and they hypothesized that the intercellular transfer of miRNAs from GBM cells to microglia might contribute to their phenotypic changes. Extracellular transfer of miR-451 and miR-21 in glioma-EVs resulted in elevated levels in microglia/macrophages and associated downregulation of their target mRNAs [72]. The exact role of EVs in glioma cell migration, invasion, and remodeling extracellular matrix ECM is still under active investigation and this offers potential actionable targets to inhibit GBM progression and invasiveness [70]. For example, Pan et al., showed that that EVs mediate the influences of glioblastoma on proliferation and migration of neural progenitors cells through the PI3K-Akt pathway, suggesting the influence of TME on normal cells [74,75]. The specific roles that EVs play in intercellular communication is due to their selective and cell-type-specific loading of lipids and nucleic acids, including DNA, messenger RNA (mRNA), microRNA (miRNA), and non-coding RNA (ncRNA) [71]. For example, miRNAs and lncRNAs packed as EV cargoes can serve as a potential mechanism driving epithelial to mesenchymal transition that involves an increase in prominent transcription factors mediating EMT such as Snail, Slug, Twist, and Zeb1, and a subsequent increase in mRNA levels of mesenchymal markers such as Vimentin and MMPs. Emerging data suggest that glioblastoma derived exosomes GBex reprogram the tumor microenvironment. GBex carried immunosuppressive proteins, including FasL, TRAIL, CTLA-4, CD39, and CD73, but contained few immunostimulatory proteins. In the study of Azambuja et al., GBex mediate extensive molecular reprogramming of all types of immune cells that leads to their functional paralysis and promotes tumor progression. GBex induced changes in the cytokine secretion profile of microglia, converting them to cells promoting GB growth and invasion while suppressing functions of immune cells [5]. One of the first studies to examine the cargo of GBM cell line-derived EVs was performed by Skog et al. [76], then several proteins in glioma cell line derived small EVs were identified. The enrichment of specific proteins and RNAs in EVs indicates that regulation and selection of cargo may take place during EV formation and release of this cargo may have an impact on gene expression levels or associated signaling pathways within the recipient cells. There are several ways in which EVs are uptaken, such as interactions between the surface of recipient cell and the EV-membrane, fusion of EVs with the plasma membrane of the recipient cell, internalization of EVs via endocytosis, and extracellular proteases which can cleave membrane proteins on the EV surface [64]. Berenguer et al. discovered a new mode of GB-EV uptake whereby glycosaminoglycans on the EV surface bind the soluble ligand CCL18 [77]. Microtubules and the actin cytoskeleton regulate the transport of exosomes. The overexpression of cortactin, a central actin cytoskeletal regulatory protein, has been shown to increase the number of small EVs released from cancer cells, revealing that targeting cytoskeletal remodeling may impact upon vesicle secretion [71].

### 3.6. Therapeutic Perspectives

Glioma stem cells (GSCs) and glioma immune microenvironment play an important role in initiating glioma and promoting tumor progression and recurrence. A lot of microglial cells infiltrate glioma tumor mass, and the degree of infiltration is positively correlated with the malignancy of such tumors. In mouse glioma models, it has been found that microglial cells are predominantly located in the margins of gliomas, and that microenvironment is associated with reduced patient survival. Current evidences are summarized in Table 1.

Glioma cells produce immunosuppressive factors such as TGF-β and IL-10, so it is difficult to use immunotherapy to treat glioma. In addition, the glioma tumor microenvironment is filled with many Tregs, normally not present in human brain tissue, and M2 tumor-associated macrophages that play an immunosuppressive role in glioma and promote tumor growth by secreting cytokines, mediating the NF-κB-IL6-STAT3 signaling pathway. Changes in the immune microenvironment could have potential therapeutic effects.

Some studies suggest the importance of complex interactions between glioma cells and their microenvironment. It is known that glioma associated microglia/macrophages (GAMs) promotes tumor growth and resistance against standard chemoradiotherapy. Resident microglia rather than peripheral macrophages promote the vascularization of brain tumors. For these reasons, many efforts have been made to try blocking the pathways that favor these tumor promoting processes.

Several GAM-derived factors such as TGF-β, stress-inducible protein (STI)-1, IL-6, IL-1β, and EGF have been identified as promoters of glioma cell invasion. Microglial cells express the membrane-spanning EGF precursor, which is capable of activating EGFR signaling in glioblastoma cell. EGFR is overexpressed or amplified in about 60% of primary glioblastoma and characteristic for a highly aggressive phenotype.

GAM facilitate the vascularization of brain tumors by secreting pro-angiogenic factors such as (VEGF). VEGFR blockage by the administration of Sunitinib (Sutent) combined with the VEGF inhibitor Bevacizumab (Avastin) resulted in a reduction in myeloid infiltrates, decreased tumor vascularity, and prolonged survival in a GBM mouse model [78].

Increased IL-10 expression levels are associated with glioma malignancy, and GAMs are regarded as the main source of IL-10 in human GBM. The transcription of IL-10 in GAMs is mainly based on STAT3 signaling. Activated pSTAT3 expression has been shown to be associated with an increased tumor grade and a worse survival of glioma patients. A siRNA-mediated STAT3 inhibition resulted in tumor growth inhibition. Some results suggest STAT3 targeting as a promising treatment approach [78].

It is known that glioblastoma (GBM) promotes M2 phenotype of microglia by secreting pro-inflammatory cytokines as IL-10, IL-13, CCL2, etc. M2 microglial cells subsequently promote GBM proliferation and survival via secretion of TGF-β and IL-6. Long non-coding RNA (lnc) molecules have an oncogenic role; one of these lnc is SNHG15. It is upregulated in GBM cells compared to the normal brain, and associated with a poor prognosis for patients. Increased lncSNHG15 level is associated with over-expression in markers of oncogenesis such as EGFR, CDK6, and Sox2. Higher levels of lncSNHG15 in TMZ-resistant clinical GBM samples was associated higher tumor aggressiveness. A preclinical study of 2019 showed the role of lncSNHG15/CDK6/miR-627 regulatory circuit in the development of GBM and polarization of GAMs both in vitro and in vivo models [79]. CDK6 has been shown to be overexpressed in GBM and is higher in temozolomide resistance cells. Treatment with CDK6 inhibitor, Palbociclib, suppressed GBM tumorigenesis and the ability to generate M2 GAM and glioma stem cells. Palbociclib’s anti-GBM effects were associated with a reduced level of lncSNHG15 and up-regulation of tumor suppressor miR-627. LncSNHG15 knockdown and palbociclib treatment both led to an increased sensitivity towards TMZ. 

Cytokines in glioma form an immunosuppressive microenvironment and hinder the production of anti-tumor immune response. Therefore, reducing or reversing this state of inhibition could be a potential target for therapies. Chemokines play an important role in the local migration of immune cells such as microglia and macrophages to glioma tumors. Chemokines ligand 1 (CXCL1) and ligand 2 (CXCL2) are closely related to tumor invasion. Targeting CXCL1/2 with standard chemotherapy can improve the chemotherapy efficiency of glioma and prolong the survival of glioma mice [80].

Glioma stem cells play a role in tumorigenesis. They can induce tumor angiogenesis, promote tumor invasion, and are highly tolerant to radiotherapy and chemotherapy. Some immunotherapies targeting GCS have been tested. In a phase one trial on mice models, NSC-CRAd-S-pk7 injection was shown to be safe and effective in patients with newly diagnosed GBM during surgery. This is a combination of oncolytic adenoviruses (CRAd-S-pk7) that target GSCs with Neural Stem Cell (NSC) that can cross blood–brain barrier. Multi-site injections in the brain improve chemoradiotherapy treatment efficiency. Another study shows that Zika virus can target GSC through SOX2 integrin axis, induce apoptosis, and inhibit proliferation of GSC [81]. In addition, glioma stem cell-related vaccines targeting SOX2 and other glioma stem cell-related genes may provide a new regimen for active immunotherapy, because SOX2 is an important transcription factor for maintaining glioma stem cells.

Research has shown that the tumor microenvironment has complex mechanisms acting as checkpoints to regulate immunosuppression. Two of the most relevant immune checkpoints are cytotoxic T lymphocyte-associated protein antigen-4 (CTLA-4) and programmed cell death-1 (PD-1). Inhibitors that block CTLA-4 and PD-1 have been approved by the FDA. Nivolumab (PD-1 antibody) combined with radiotherapy, chemotherapy, and electric field therapy can improve the survival of some glioma patients [82].

Microglia account for 5–10% of total brain cells, and its physiological role is maintaining homeostasis of the central nervous system. In gliomas, this proportion increases significantly, even up to more than 30% of tumor mass. These cells interfere with tumor progression. Therefore, microenvironment changes promote the growth and migration of cancer cells. Some of these events are mediated by glutamate. A recent study on glioma bearing mice demonstrated the contribution of microglia derived small extracellular vesicles (sEV) transporting miR-124 to maintain brain homeostasis of glutamate balance in extracellular space [83]. MiR-124 increase the expression of Glutamate transporters, reducing the release by tumor cells. This downregulation protects neurons from glutamate induced neurotoxicity mediated by glioma cells. The upregulation of miR-124 reduce glioma cell proliferation and angiogenesis, and its overexpression increased survival in a mouse model. Microglia derived sEVs are stable ex vivo for months and are easy to produce. These characteristics make them a potentially applicable therapeutic strategy.

GAMs are known to support glioma cells, fostering tumor progression and immune suppression and treatment resistance by secreting cytokines and growth factors. GAMs were found to be involved in the radioresistence of glioblastoma by secretion of TNF α, which increases nuclear factor-κB (NF-κB) which is related to inferior survival. Many strategies have been considered to reduce the recruitment of GAMs in glioma progression, modulating GAMs phenotype toward a more anti-tumor M1-like microenvironment, or decreasing M2-like tumor promoting microenvironment. For example, inhibition of NF-κB signaling leads to M1-like TAM polarization in mouse models of glioblastoma and other tumors, such as prostate and pancreas, which leads to increased infiltration of cytotoxic T cells and decreased tumor growth [84].

Toll-like receptors (TLRs) are in glial cell, immune cells, and neurons. They modulate immune responses in innate immune system and in glioma, suppressing pro-tumorigenic pathways. Thus, they could represent potential new target in glioma therapy. An example is TLR3 agonist: polyinosinic–poly-cytidylic acid and poly-L-lysine (Poly ICLC–Hiltonol-) which are a double-stranded RNA complex able to activate immune cells working as vaccine adjuvant, favoring migration of immune cells into tumor mass. In a phase I trial, newly diagnosed GBM patients were administrated Poly ICLC along with standard care (chemoradiation) and personalized peptides based on individual tumor mutation as tumor-specific antigens vaccination. Poly ICLC was tested, together with other immunostimulants such as TLR3 agonist Poly I:C and TLR7/8 agonist imiquimod (Aldara), in cancer vaccine therapies on patients undergoing complete tumor resection of high grade gliomas. Poly I:C/Poly- ICLC was injected intramuscularly or subcutaneously, while imiquimod cream was administrated on skin before and/or after each vaccination. An increase in PFS and OS was observed in both studies, giving hope for useful future therapies [85].

## 4. Conclusions

All this evidence supports the idea that the genesis, growth, and progression of gliomas are not exclusively dependent on the intrinsic mutations of neoplastic cells, but are strongly influenced by the surrounding microenvironment and by the communication between tumoral and non-tumoral cells. Hence, it appears essential to strengthen the focus of scientific research on the study and the understanding of GAM, in order to identify new therapeutic targets and treatment strategies for glioma patients. In particular, next generation approaches to EV analysis will allow the full potential of EVs as glioma biomarkers to be realized. However, no significant results have yet been obtained in studies targeting single molecules/pathways. It is likely that combining different treatments could have an advantage. The association of chemoradiotherapy plus GAMs target therapies could reprogram TAMs, improving standard treatment effects. Surely, it is paramount to increase our knowledge about TAM subpopulation and the complex interaction with glioma cells. Given the existence of such a structured microenvironment, it is likely that only by using multiple therapeutic agents acting on multiple molecular targets can we obtain significant results in the fight against glioma.

## Figures and Tables

**Figure 1 ijms-23-04166-f001:**
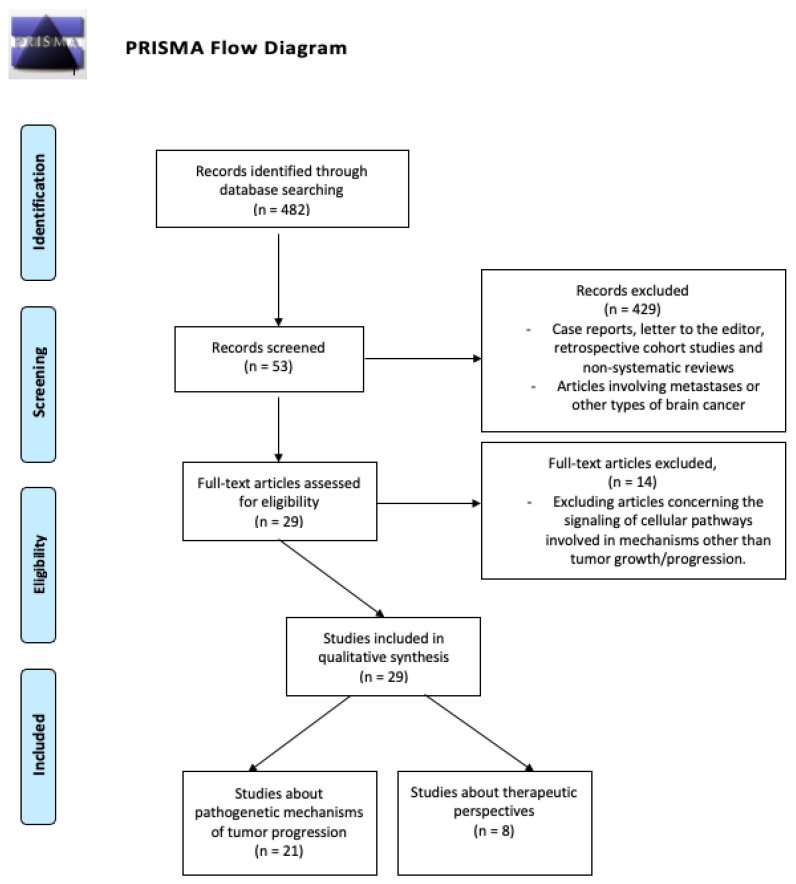
Flow diagram showing the article selection process.

**Figure 2 ijms-23-04166-f002:**
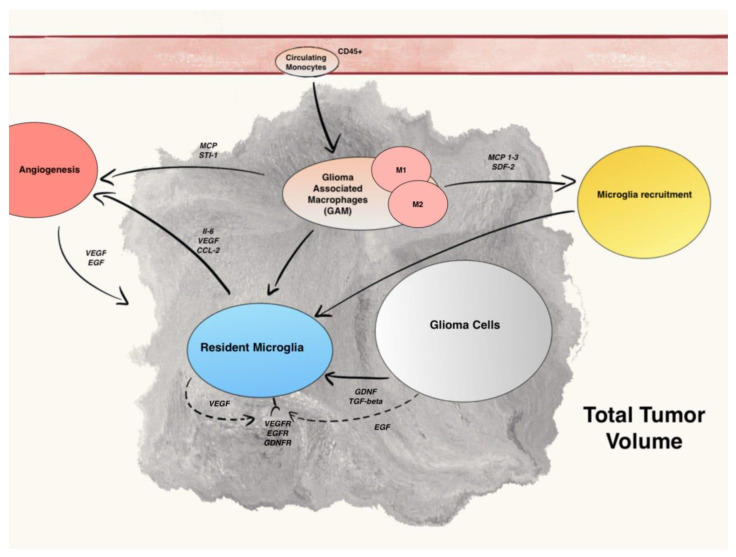
Summary of main communication pathways between GAM, microglia, and Glioma cells.

**Figure 3 ijms-23-04166-f003:**
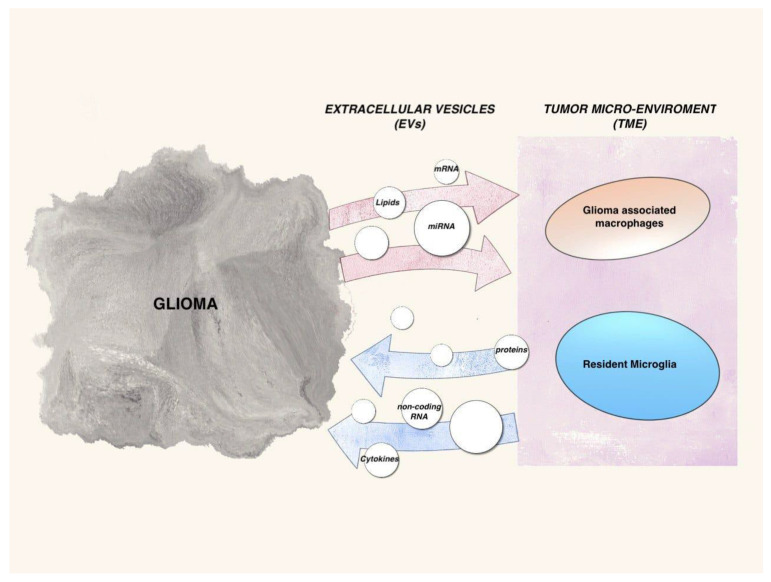
Schematic representation of glioma-TME communication through EVs.

**Table 1 ijms-23-04166-t001:** Literature synthesis on therapeutic perspectives.

Author, Year	Type of Article	Main Pathway Involved	Specific Target Molecules	Drug Tested	Human/ Mice/Vitro	Results
***Roesch*** **et al., *2018***	Review	GAMs	VEGF-VEGFR	Sunitinib + Bevacizumab	Mice	Increased OS
			IL-10, STAT3	siRNA mediated STAT3 inibitor	In vitro	Tumor growth inhibition
***Li*** **et al., *2019***	Research	GAMs	CDK6	palbociclib	In vitro	Increased TMZ sensitivity
** *Hu, 2021* **	Research	Cytokines	CXCL1/2	CXCL1/2 inhibitor + TMZ	Mice	Combination of targeting CXCL1/2 and TMZ increased OS
***Li*** **et al., *2021***	Review	GSCs		NSC-CRAd-S-pk7	Mice	Improved efficacy of chemoradiotherapy treatment
			SOX2	Zika Virus	In Vitro	Apoptosis of GSCs
***Chadran*** **et al., *2017***	Review	Tumor associated Lymphocytes	CTLA4-PD1	CTLA4-PD1 inhibitors (eg. Ipilimumab)	In vitro	Improve OS in some gliomas in combination with chemoradiotherapy
***Serpe*** **et al., *2021***	Research	EVs	Glutamate, miR-124	miR-124 upregulation	Both in vitro and mice	Tumor mass reduction in vitro and increased OS in mice
***Andersen* et al., *2021***	Review	GAMs	TNF α- NF-κB	inhibition of NF-κB signaling	Both in vitro and mice	Increased infiltration of cytotoxic T cells and decreased tumor growth.
***Xun*** **et al., *2021***	Review	TLRs	TLR3	Hiltonol (TL3 agonist)	Mice	Increase in PFS and OS
			TLR7/8	Imiquimod		

## Data Availability

Not applicable.

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
