# Peer review of "Systematic Review on Tumor Microenvironment in Glial Neoplasm: From Understanding Pathogenesis to Future Therapeutic Perspectives"

_ijms, 2022, doi:10.3390/ijms23084166_

Round 1

Reviewer 1 Report

In the present manuscript, Bianconi et al., present and interesting systemic review on glioma associated microglia. The systematic review methodology is sufficiently described and comprehensive. The authors provide nice evidence of different molecular targets in microglia during gliomagenesis with a particular emphasis on extracellular vehicles. However, I am surprised that their systematics did not pick up two key publications describing deep sequencing microglia analysis in human glioma. I would appreciate an adjustment of the methodology given the high impact of these two publications. Moreover, the lack of graphical illustrations is hampering the reading experience. The manuscript would greatly benefit from conclusive graphics at the end of each paragraph.

Mayor points:

I am missing the inclusion of two key publications:

Mapping microglia states in the human brain through the integration of high-dimensional techniques. Sankowski et al. 2019 Nature Neuroscience

Single-Cell Mapping of Human Brain Cancer Reveals Tumor-Specific Instruction of Tissue-Invading Leukocytes. Friebel et al. 2020 Cell

Regarding these two studies please revisit the M1/M2 polarization phenotype critically. This approach might be outdated given the large screening capacities available today.

I would appreciate concluding illustrations after each paragraph.

Minor points:

Please cite always at the end of a sentence.

Line 292 reference is missing.

Line 366 (Siglech) ??

Author Response

Mayor points:

Point 1:

I am missing the inclusion of two key publications:

Mapping microglia states in the human brain through the integration of high-dimensional techniques. Sankowski et al. 2019 Nature Neuroscience

Single-Cell Mapping of Human Brain Cancer Reveals Tumor-Specific Instruction of Tissue-Invading Leukocytes. Friebel et al. 2020 Cell

Regarding these two studies please revisit the M1/M2 polarization phenotype critically. This approach might be outdated given the large screening capacities available today.

Response 1: 

Unfortunately the two cited articles were not included in the systematic review as they were automatically excluded from the search string. Recognizing the undoubted value, they have been added and commented, with particular regard to the M1-M2 polarization. These articles in fact confirm what was stated in our review on the fact that it is an outdated concept, since more specific screening techniques are available.

Point 2:

I would appreciate concluding illustrations after each paragraph.

Response 2:

Two summary images were added to the microglia and extracellular vesicle paragraphs, respectively.

Point 3:

Minor points:

Please cite always at the end of a sentence.

Line 292 reference is missing.

Line 366 (Siglech) ??

Response 3:

Minor revisions were made as required

Reviewer 2 Report

Dear Authors,

In the review article entitled „Systematic review on tumor microenviroment in glial neoplasm: from understanding pathogenesis to future therapeutic perspectives” the authors aimed to review the current state-of-the-art research in glioblastoma, with special emphasis on the role of extracellular vesicles tumour associated macrophage and microglial cells.  Overall I consider that, the authors could better elaborate on role of EVs in glioblastoma.

I have the following major points:

(1) Microglial, macrophage phenotype by different CD markers/ intracellular molecular pattern should be presented in a figure, therefore the reading on the manuscript would be significantly improved.

(2) In the characterization of tumor associated microglia and macrophages there should be a discrimination between murine and human in vitro cell line models, as the phenotype of human and murine macrophage do not completely overlap.

 (3) Role of EVs in glioblastoma context should include more relevant articles as: PMID: 34902452, 33508126, 35052830, 35022057

I have the following minor points:

(1) “in vitro” and “in vivo” should be italicized when used: “in vitro”, “in vivo” e.g. LINES 59, 137, 139, 152, 163, 165, 174, 252, 265, 271, 282, 305, 354, 457,

(2) LINE 114  “antiCD45 antibody” should be corrected to “anti-CD45 antibody”

(3) LINES 131, 172, 175, 192, 433  I would suggest either the use of “microglial cells” or “microglia” without “cells”

(4) LINE 132 “neurotransmitters” instead of “neurotransmitter”

(5) LINE 157 at first use interleukin 1β should be written fully accompanied by the abbreviation (IL-1 β)

(6) LINE 159 the authors should be more specific on “human models” – are these studies performed on in vitro experiments, 3D Matrigel models, ex vivo implants?

(7) LINE 193 “cluster of differentiation CD” should be abbreviated upon first use in line CD114

(8) LINE 222 “Il-10” should be corrected to “IL-10”

(9) LINE 241 “alternative name colony-stimulating factor 1 (CSF-1)” should be excluded from the manuscript, as it is not relevant here

(10) LINE 265 a space is missing “M1/M2model” „M1/M2 model”

(11) LINES 274-275 reference 48 should be inserted at the end of the sentence

(12)  LINE 292 reference is missing

(13) LINE 340 there is a misspelling „membrane” instead of „membran”

(14) LINE 361 TNF-a and IL-6 have already been abbreviated, the short form should be written, instead of writing the long form.

(15) LINE 422 regulatory T cells have already been abbreviated in LINE 214

(16) LINE 438 VEGF, VEGFR has been previously abbreviated before, please use here the abbreviated form

Kind regards,

Author Response

Point 1

(1) Microglial, macrophage phenotype by different CD markers/ intracellular molecular pattern should be presented in a figure, therefore the reading on the manuscript would be significantly improved.

Response 1

1) a summary figure has been added concerning the main molecular pathways through which microglia, glial cells and macrophages communicate, to make reading the article more usable

Point 2:

(2) In the characterization of tumor associated microglia and macrophages there should be a discrimination between murine and human in vitro cell line models, as the phenotype of human and murine macrophage do not completely overlap.

Response 2:

2) We agree that the two cell lines have different characteristics and are reported separately, as in the paragraph microglia recruitment factors.in the paragraph concerning microglia and macrophages, murine and human in vitro cell line models have been more clearly distinguished, underlining the differences. 

Point 3

(3) Role of EVs in glioblastoma context should include more relevant articles as: PMID: 34902452, 33508126, 35052830, 35022057

Response 3

3) the requested articles have been added. Since these are very recent articles published in late 2021 or 2022, the reason for their lack of citation is that they were not yet available when the search string was prepared.

Minor revisions

All the 16 minor revisions were made as requested

Round 2

Reviewer 1 Report

All my questions have been addressed

Reviewer 2 Report

The authors have addressed my main questions, I consider that the paper is suitable for publication.

Kind regards,